# Etiopathogenesis, Diagnosis, and Treatment Strategies for Lymphomatoid Papulosis with Particular Emphasis on the Role of the Immune System

**DOI:** 10.3390/cells11223697

**Published:** 2022-11-21

**Authors:** Danuta Nowicka, Paulina Mertowska, Sebastian Mertowski, Anna Hymos, Alicja Forma, Adam Michalski, Izabela Morawska, Rafał Hrynkiewicz, Paulina Niedźwiedzka-Rystwej, Ewelina Grywalska

**Affiliations:** 1Department of Dermatology, Venereology and Allergology, Wrocław Medical University, 50-368 Wrocław, Poland; 2Department of Experimental Immunology, Medical University of Lublin, 20-093 Lublin, Poland; 3Department of Forensic Medicine, Medical University of Lublin, 20-090 Lublin, Poland; 4Department of Clinical Immunology and Immunotherapy, Medical University of Lublin, 20-093 Lublin, Poland; 5Institute of Biology, University of Szczecin, 71-412 Szczecin, Poland

**Keywords:** lymphoproliferative disorder, immune system, dermatology, CD30, treatment, diagnosis, histopathology, immunophenotyping

## Abstract

Lymphomatoid papulosis (LyP) is a very rare disease that belongs to the group of CD30+ lymphoproliferative skin diseases. LyP is localized or generalized and usually presents as isolated or clustered red/brown-red lesions in the form of nodules and/or papules. The course of the disease is in most cases mild; however, depending on concomitant risk factors and history, it may progress to lymphoma, significantly reducing the survival rate and prognosis. Importantly, the clinical picture of the disease remains somewhat ambiguous, leading to a large number of misdiagnoses that result in inappropriate treatment, which is usually insufficient to alleviate symptoms. In addition to clinical manifestations, the histological characteristics vary widely and usually overlap with other conditions, especially those belonging to the group of lymphoproliferative disorders. Although diagnosis remains a challenge, several recommendations and guidelines have been introduced to standardize and facilitate the diagnostic process. This article reviews the available literature on the most important aspects of etiopathogenesis, clinical and histopathological features, diagnostic criteria, and possible treatment strategies for LyP, with particular emphasis on the role of the immune system.

## 1. Introduction

Lymphomatoid papulosis (LyP) is a relatively rare disease, occurring in the range of 1.2 to 1.9 cases per million people per year [1]. The data in the literature show that it is characterized by a bimodal peak incidence, the highest of which occurs in the 4th and 5th decade of life, especially in women, while the second, smaller peak occurs in children up to 18 years of age, with a higher percentage of male patients [2]. LyP presents with lesions that look like small red or red-brown bumps only a few millimeters in diameter, or they may appear as spots on the skin. As the disease progresses, the lesions may evolve into larger nodules and/or plaques and/or papules, usually with a maximum diameter of 2 cm, or they may go into spontaneous remission [3]. The etiology and pathogenesis of LyP remains unclear and is the subject of many interdisciplinary studies, including the mechanisms of spontaneous remission of the disease. Scientists are looking for evidence of involvement in the etiopathogenesis of onocogenic viruses (EBV, herpes virus), the participation of atopy (observed in about 50% of patients), genetic factors and susceptibility (aneuploidy and chromosomal aberrations), and abnormalities in the immune system [4,5]. 

According to the World Health Organization (WHO), LyP belongs to a family of diseases called CD30+ primary lymphoproliferative diseases of the skin (pcCD30 + LPD) [6]. This means that, in its course, there is an uncontrolled production of lymphocytes (T lymphocytes expressing CD30+) that contribute to the disorder of the immune system and the loss of immune homeostasis [7]. CD30 is a surface cytokine receptor belonging to the superfamily 8 of tumor necrosis factor (TNF) receptors, which is expressed on activated T and B cells. The interaction of CD30 with its membrane-bound glycoprotein ligand (CD30L) activates the kappa-B (NFkB) nuclear signal transduction pathway), causing either cell proliferation or apoptosis. Due to CD30 suppression in LyP, which is also observed in some hematopoietic system neoplasms (primary cutaneous anaplastic large cell lymphoma (ALCL) and Hodgkin’s lymphoma (HD)) [8,9], this disease can be viewed as benign lymphoma, but also as a precursor to cutaneous T-cell lymphoma (CTCL), although some experts consider it to be a very low-grade form of CTCL [10,11]. Contrary to the fairly consistent clinical picture, the histological classification of LyP covers a broad spectrum of subtypes (A, B, C, D, E, and a subtype with posterior DUSP22-IRF4) [12,13,14], resulting in a difficult diagnostic process, ultimately leading to a large number of misdiagnoses and inadequate therapies that delay recovery and may even affect patient survival [15]. 

Due to the overexpression of CD30 and the similarity to inflammatory diseases, it is extremely important to make a correct diagnosis to perform a differential diagnosis in order to exclude diseases such as HD, lichen dandruff, primary cutaneous ALCL, mycosis fungoides (MF), CTCL, and other mild conditions such as atopic dermatitis, viral infections, scabies, mycobacterial infections, and drug reactions [12,15,16,17,18]. Therefore, the aim of this review was to approximate the most important aspects related to etiopathogenesis, clinical and histopathological characteristics, diagnostic criteria, and possible treatment strategies that may be useful tools in the diagnosis and selection of therapeutic strategies in patients with LyP. Additionally, the important role of the immune system in the development of this disease is discussed.

## 2. Epidemiology of Lyp in Terms of Histological Types

Currently, LyP may represent up to 12% of all diagnosed skin lymphomas [19,20]. It is the most frequently diagnosed skin lymphoma in the fourth and fifth decade of life in all ethnic groups equally [21,22]. The most important risk factor for LyP is HD or CTCL or a medical history of these two conditions, while the most common lymphomas associated with LyP are MF (24–61.4%), primary cutaneous ALCL (13–44%), and HD [21,23,24,25,26]. The risk of developing LyP-associated lymphoma is 2 to 7.5 times higher in patients with the occurrence of a monoclonal rearrangement of the TCR-γ (T-cell receptor) gene chain in skin lesions [26,27]. LyP has a diversified clinical and histological picture; while the misdiagnosis rate is difficult to determine, it is estimated to be about 30%, frequently resulting in unnecessary antibiotic treatment, chemotherapy, or radiotherapy [21,25]. LyP type A is the most diagnosed, with an estimated incidence of 47.2–82% compared to the other subtypes [12,21,26,28]. It has a 5-year survival of 100% [5,26,29]; however, the prognosis may be reduced by the increased risk for progression to lymphomas, which is estimated to be 10–20% for LyP with adult-onset and 10% in the pediatric population [23,26].

## 3. Clinical Manifestation in LyP

Molecular and histological aspects must be discussed together with clinical presentation. The genetic instability of tumor cells is responsible for several characteristics. First, patients must be monitored for the development of lymphoma. Second, a patient may present with more than one histological subtype of LyP. Finally, the disease can progress or resolve spontaneously. All patients with LyP require long-term control visits, twice a year, to evaluate their clinical presentation, as described below.

LyP is a chronic disease of cutaneous lymphoid infiltration, characterized by a diverse clinical morphology and the occurrence of skin lesions such as papules, plaques, and nodules [30]. Early lesions appear as small red or reddish-brown or red-violet nodules a few millimeters in diameter that may be singular, clustered, or generalized [31]. As they grow, these lesions can develop into larger nodules and plaques, usually with a maximum diameter of no more than 1–2 cm [32,33,34]. Although complete regression may occur within a few weeks, papules may also progress to sterile pustules or become necrotic and then lead to hemorrhagic scabs and varioliform atrophic scarring [35,36]. In patients with intermittent LyP ejection, changes may coexist in different stages of development, resulting in a differentiated and polymorphic clinical picture [15]. Skin lesions can occur anywhere on the body; however, they usually develop mainly on the limbs and torso and, less often, on the face [37]. There are only a few reports of oral or genital involvement in the literature [38,39]. 

A characteristic feature of skin lesions and an important diagnostic criterion of LyP is the potential for spontaneous regression. These changes can resolve on their own within a few weeks to a month from onset (this time is estimated at 3 to 12 weeks, depending on the patient). In most patients, skin lesions do not cause additional symptoms, but some report that the lesions are accompanied by itching and pain (approximately 40–55% of patients). The clinical features of LyP do not include palpable lymph nodes or enlargement of the liver and spleen [40,41]. In terms of morphology, skin lesions present an extremely diverse manifestation. Basic nodular lesions can include up to several hundred lesions in the trunk and/or limbs, which, over time, turn into post-inflammatory lesions, appearing as discolored spots. Ultimately, they transform into atrophic varicose scars in the spontaneous resolution [41]. In addition to common nodules, skin lesions can also take the form of diffuse, serrated, and lamellar lesions [42,43]. There are also less common morphological variants of LyP that include vesicular, eczema, or lamellar ulcerative symptoms [32,33,34,44].

Over the course of the disease (or the relapse of skin lesions), the morphology of skin lesions also changes. Initially, an inflammatory papule is formed, accompanied by tortuous and irregular vessels radiating from the inside. A fine, dotted, vascular pattern surrounds a central homogeneous pink-light brown area. A mature hyperkeratotic papule then forms that lacks a central vascular component that can persist at the periphery. Central scales or crusts and peripheral rim resemble porokeratoris. In the next stage, a central necrotic ulcer or crust appears, accompanied by a brownish-gray unstructured area. Peripheral vessels may be visible. In the last phase, called the scarring phase (cicatricial phase), a brown spot without vasculature appears [18]. Lesions that appear as lymphomatoid papulosis are shown in Figure 1.

## 4. Etiopathogenesis and Role of the Immune System in the Development of LyP

In general, the pathomechanism and etiology of LyP remain unclear. However, LyP is classified among primary cutaneous CD30-positive lymphoproliferative disorders, which represent the second-most common group of CTCL [46]. This background elucidates the shared pathological mechanisms that explain the etiology of LyP. The main aspect of CTCL etiology is persistent antigenic stimulation [46,47]. So far, there is no evidence to suggest that LyP may be the result of bacterial, viral, fungal, or any other type of infection. Nonetheless, there are a few reports in the literature in which some authors suggested that LyP may be the result of viral infections induced by HTLV-1, herpesviruses, or endogenous retroviruses. Furthermore, given the distribution of patients with CTCL and their clustering in certain locations, e.g., heavy industrial factories, major transportation hubs, or sunny desert climates, some exposures may serve as factors promoting this cancer [48]. However, current data are not sufficient to unambiguously confirm these hypotheses [47,49]. 

According to available clinical data, CTCL is derived from CD4+ CD45RO+ T cells, which are abundantly present in the skin [50,51]. Genetic factors predisposed to LyP are also still unknown; however, HLA class II alleles or mutations within the transforming growth factor-β type I receptor have been found to be potentially involved in the pathogenesis of CTCL [52,53,54,55]. It has been suggested that there may be a relationship between the etiology of LyP and primary cutaneous ALCL t (2;5) (p23, q35), as LyP cells have been reported to present the gene for the oncogenic transcription factor Fra2 together with the differentiation inhibitor Id2 gene, both located in close association with t (2;5) [56,57]. Interestingly, a gene fusion (NPM1-TYK2) was found in cell biopsy samples from patients with LyP and primary cutaneous ALCL, once again showing a possible relationship between the two conditions [58,59]. Additionally, approximately 40%, even up to 100%, of LyP patients show clonal rearrangement of TCR genes [60,61]. Some of the aforementioned aspects include CTCL in particular, and although LyP is a subtype of it, the pathomechanism described may not be fully applicable to this condition, or there may be many more that interact but remain undiscovered.

## 5. Importance of Histopathological Immunophenotyping and Immunohistochemistry in the Diagnosis of LyP

LyP belongs to the lymphoproliferative disorders of the skin characterized by excessive production of CD30+ cells, but it is not a homogeneous syndrome, the diagnosis of which requires many clinical, histological, and immunohistochemical tests [62,63,64,65]. The first and most important step in establishing a diagnosis of LyP is histological examination. Complete excision of skin lesions, incision biopsy, and punch biopsy are sufficient and provide biological material for a reliable histological evaluation [62]. Studies have shown that skin lesions of T lymphocytes expressing the CD30+ receptor most often show a specific cytotoxic phenotype. Typically, these are CD30+/CD4+/CD8− cells, expressing both granzyme B and intracellular antigen confined to T lymphocytes [13,66]. Particular features of LyP subtypes can be combined and appear together in one patient: each lesion may have a different histological subtype [67]. The difficulties in the diagnosis of this disease may result from histological similarity to other diseases in the spectrum of primary and secondary cutaneous CD30+ lymphoproliferation [68,69]. In the diagnostic process, the technique of immunophenotyping conducted with the use of flow cytometry (FCM) is increasingly being used, which allows a more precise determination of the individual types of cells present in the tested material. This is possible because of the identification of individual cells on the basis of a combination of physical characteristics and the use of multiple antibodies directly conjugated to different fluorochromes. In addition, flow cytometric immunophenotyping (FCI) has become a widely used laboratory procedure for diagnosing and identifying subtypes of many diseases, including lymphoma. Immunophenotyping can also be performed on fresh peripheral blood/bone marrow/lymph node aspirate samples by FCM or on formalin-fixed paraffin-embedded tissue by immunohistochemistry (IHC). Each of the available methods has its advantages and certain limitations and offer an extremely important contribution to the final diagnosis [70,71].

### 5.1. Type A LyP

Type A is the most common subtype of LyP (approximately 75% of cases) and is characterized by a wedge-shaped infiltration of diffuse or clustered large atypical CD30+ cells, interspersed with numerous inflammatory cells such as small lymphocytes, neutrophils, eosinophils, and histiocytes. Furthermore, in this type, atypical cells represent less than 50% of infiltrating cells and do not form sheets [72]. Type A resembles polymorphic infiltrates of Hodgkin’s lymphoma. Regarding immunophenotyping, type A is characterized by the expression of CD30+ and CD3+ on pathogenic lymphocytes. A characteristic wedge-shaped infiltrate is observed, composed of pleomorphic cells with histiocytic-like characteristics similar to Reed–Sternberg cells, which are CD30+ and are interspersed with eosinophils, neutrophils, and small lymphocytes. As for the CD8+ and CD4+ antigens, some penetrating cells are double-negative and some double-positive, while some express mainly one of those with a different frequency. Typically, CD4+ is dominant in CD30+ lymphoproliferative diseases such as LyP [73]. Furthermore, the participation of other markers, such as CD2, CD5, and CD7 (whose expression is differentiated and, depending on the sample, may be positive or negative) have been reported in the course of LyP [74] (Table 1). One possible finding of note is the expression of TIA-1 (T cell intracellular antigen 1, cytotoxic granule-associated RNA binding protein) found on cytotoxic T lymphocytes, which is associated with programmed cell death (apoptosis) and is responsible for the regulation of alternative splicing of the Fas receptor gene [75].

### 5.2. Type B LyP

Type B lesions are dominated by epidermotropic infiltration of smaller, atypical CD30+ or CD30− cells with cerebriform nuclei that histologically resemble MF. These infiltrates are deep and reach even the reticular layer. Epidermotropism and epidermal hyperplasia are very common in type B LyP. In this type, mitotic figures are rarely described [67]. In contrast to MF, type B LyP infiltrates are associated with follicular lesions that resolve spontaneously [36]. Type B is characterized by the immunophenotype CD30−, CD3+, CD4+, and CD7− [76]. Less frequently than in type A, lymphocytes show CD8+ expression or do not show it at all. Cells also show differential expression of CD2 and CD5 similar to type A [13] (Table 1). This type also includes cells with a clonally rearranged TCR receptor and those expressing TIA-1 [67].

### 5.3. Type C LyP

Type C lesions (similar to those of type A) contain large clusters or sheets of large atypical CD30+ cells in the skin, but with relatively few inflammatory cells. These infiltrating CD30+ cells, unlike type A, account for more than 50% of the infiltrating cells by histology. In type C, superficial and deep skin infiltration with epidermal hyperplasia is observed [67]. As in type A, lymphocytes can be multinucleated, and numerous mitotic figures are observed. These changes resemble skin lesions of primary cutaneous ALCL, and the distinction from primary cutaneous ALCL is largely based on the clinical picture [36,69]. Type C immunohistochemistry closely resembles type A LyP. CD30+ cells are also present, with different phenotypic configurations in terms of CD4 and CD8 expression, but typically represent the CD30+, CD3+, CD4+ phenotype without signs of ongoing inflammation [63,65]. Expression within the TIA-1 and TCR rearrangements differs between patients [51]. Cells are also characterized by differential expression of CD2 and CD5, similar to type A [13] (Table 1).

### 5.4. Type D LyP

D-type lesions (similar to CD8+ T cell cytotoxic lymphoma) are characterized by pagetoid infiltration of small-to-medium atypical CD8+ and CD30+ epidermal cells that resemble primary cutaneous aggressive epidermal CD8+ T cell cytotoxic lymphoma. However, clinically, changes appear and behave similarly to those of LyP. In most cases, these abnormal cells account for more than 50% of the infiltrating cells. Histological examination revealed spongiosis, parakeratosis, and epidermotropism [77]. Epidermotropism is observed within the vacuole changes in the dermal–epidermal junctions [78]. The infiltrates observed in this disease are wedge-shaped, usually do not reach the subcutaneous tissue, and infiltrating cells show only moderate atypia and are not very differentiated from each other [73]. In addition, moderate epidermal hyperplasia is observed. Large abnormal cells are usually mixed with small lymphocytes of this type of LyP; large cells tended to be mononuclear, while small lymphocytes showed testicular atypicality [78]. Typical D-type LyP is characterized by an infiltrating cytotoxic phenotype of cells expressing CD30+, CD3+, CD8+, and granzyme, both in large and small cells [63]. Regarding the molecules CD4, CD5, CD7, and CD62L, in most cases, we did not observe their expression in the analyzed samples [78] (Table 1). The research conducted by the Bertolotti team in 2013 reported that in samples taken from patients diagnosed with type D LyP, no expression of the CD56 molecule was observed [79]. Furthermore, infiltrating cells also express not only the granzyme, but also perforin and the intracellular antigen-1 of T cells [79].

### 5.5. Type E LyP

Type E LyP is characterized by clinically ulcerative scab-like lesions, usually with hemorrhagic necrosis that resolves with scarring after 3–6 weeks. Histologically, the most important and typical finding for this subgroup was cellular infiltration in blood vessels. The vast majority of cells found during immunohistochemical staining in material collected from persons suspected of type E LyP were T cells with expression of the CD30 receptor and a cytotoxic phenotype (CD8+/TIA-1) [64]. These cells dominated the vascular infiltrates, although CD4+ expressing cells were also sometimes involved [80,81]. Typically, these CD30+/CD8+ lymphocytes also express CD2 and CD5. According to data from the literature, several cases of cells expressing CD30 and CD56 among cells infiltrating blood vessels have also been described [13,82]. Due to infiltration of cytotoxic cells, this may suggest fatal acute cutaneous lymphomas, therefore, the differential diagnosis should particularly include extranodal T/NK lymphoma and γ/δ-positive lymphoma to avoid misdiagnosis [65,83,84,85]. In the diagnostic material, Kempf et al. described the presence of pleomorphic lymph cells of various sizes, ranging from small to large, usually medium in size. Infiltration of medium veins and small arterioles in both dermis and subcutaneous tissue most likely led to necrosis in the surrounding ischemic tissues. Vasculitis with fibrin deposition in the vessel walls was described, and vascular thrombosis occurred in half of the cases. Accordingly, histological examination showed signs of necrosis of the skin, subcutaneous tissue, and appendages. Extravasated red blood cells were also typically observed in examined tissues [12]. Few descriptions of this rare disease are available in the literature, but more frequent searches for this disease in patients with skin lesions have been observed in recent years [73,80,86,87].

### 5.6. Type with Rearrangement 6p25.3

This variant has a characteristic biphasic histology with small and medium epidermotropic brain lymphocytes and large pleomorphic skin lymphocytes. CD30 staining is two-phase, with the most intense staining of the skin compared to epidermal cells. Atypical cells are often double negative for CD4 and CD8 or CD8+ [88,89,90].

## 6. Diagnostic Procedure of LyP with Particular Emphasis on Differential Diagnosis and Underlying Histological Features

The diagnosis of LyP remains one of the major clinical challenges in the field of lymphoproliferative disorders due to a large number of conditions with similar pathological changes and symptoms. The very first step of correct diagnostic procedure requires a complete physical examination, with a particular emphasis on skin examination. It is essential to take a detailed medical history of the patient to ensure that presented symptoms are not a recurrence of a previously unrecognized disease. A medical history of previous lesions (nodules and tumors) on the skin, HIV infection, a history of lymphoma, or immunosuppressive therapy significantly increase the risk of LyP. There are three main features that significantly increase the likelihood of a LyP diagnosis [91]:(a)Overexpression of CD30 + T cells found during immunohistochemistry;(b)Infiltration of large atypical T cells (along with numerous other inflammatory cells including small lymphocytes, neutrophils, histiocytes, and eosinophils;(c)Clonal rearrangement of TCR genes (found in approximately 40–100% of cases).

The presence of red-brown papules or nodules is an alarming signal and requires further investigation and differential diagnosis. A skin biopsy must be performed for a thorough pathological assessment; this includes immunohistochemical and histopathological evaluation, as well as genetic molecular evaluation. It should be remembered that the final diagnosis must ultimately be confirmed by a qualified pathologist [18]. Histopathological features are also strongly differentiated depending on the LyP subtype [63]. Importantly, in order to make a reliable diagnosis, the pathological examination should also include an immunohistochemical examination along with the T-cell gene rearrangement test. After showing a significantly increased number of CD30+ T cells, the pathologist must look for differences to indicate a specific subtype of this condition. Blood samples should be taken from each patient in order to properly assess the hematologic and lymphatic systems. Complete blood counts, biochemistry, and peripheral blood tests are required. In cases of visibly enlarged and palpable lymphadenopathy, imaging tests, such as computed tomography or positron emission tomography, are recommended [19] (Figure 2).

### Differential Diagnosis of LyP

The correct diagnosis of LyP is based on the correlation of clinical and histopathological results of patients. However, due to many subtypes of this disease and the similarity of clinical features with other lymphoproliferative diseases, as well as inflammatory or infectious diseases, the diagnostic process itself is significantly more difficult [92]. Most often, diagnostics are conducted to exclude lymphoproliferative diseases expressing CD30+, the vast majority of which are various types of lymphomas (primary cutaneous anaplastic large cell lymphoma, secondary skin lesions of systemic anaplastic large cell lymphoma, or Hodgkin lymphoma) (Figure 3) [93,94,95,96].

Diagnostics to rule out the presence of lymphomas should include an analysis of the medical history and risk factors (history of lymphocytic neoplasms; presence of systemic symptoms such as unexplained weight loss, fever, night sweats, dyspnoea, or abdominal fullness; HIV infection or previous immunosuppressive treatment), detailed analysis of skin lesions (morphology; size and extent of skin lesions; occurrence of erythematous; scaly spots and plaques; enlargement of the liver or spleen), as well as analysis of laboratory tests (complete blood count with differentiation for atypical cells and biochemicals, including lactate dehydrogenase) [62]. Detailed differential diagnosis for specific lymphoma subtypes should include primary cutaneous anaplastic large-cell lymphoma (PC-ALCL), systemic anaplastic large-cell lymphomas (ALCLs), transformed mycosis fungoides (MF), adult T cell leukemia–lymphoma (ATLL), and Hodgkin’s lymphoma. 

In both first diseases, CD30+ overexpression was observed, which occurred in 75% of cases, and clonal TCR rearrangement in almost 90% of cases. In a study conducted by the Humme team in 2009, it was shown that the monoclonal rearrangement of the TCR receptor genes occurs in 78% of skin samples and 36% in blood samples from LyP patients [97]. Detailed analyses of the clonal populations in both types of samples showed a different pattern of rearrangement. The researchers suggested that the T-cell clones that were observed in the peripheral blood did not show neoplastic features and could be a kind of response to an unknown antigen [97]. Moreover, in the studies available in the literature on PC-ALCL, researchers did not observe expression of epithelial membrane antigen (EMA) or anaplastic lymphoma kinase (ALK), which are present in systemic ALCL [93,94,95]. The third disease subunit is the greatest diagnostic challenge in relation to type B LyP, in which, apart from CD30+ overexpression, MF atherosclerotic plaque is observed [98].

At the moment, there is no information on studies detailing the differentiation of these two diseases. Another disease for which differentiation should be made in the diagnosis of LyP is adult T cell leukemia–lymphoma (ATLL). This is a disease caused by the human HTLV-1 retrovirus (human T-cell leukemia virus type 1), which begins with infection in the skin in almost 50% of cases. Very often, the altered skin cells express CD30+ while being ALK-, which combined with the presence of HTLV-1 antibodies in peripheral blood, allows distinction of this disease entity from LyP [99]. In Hodgkin’s lymphoma, diagnosis is performed to detect Reed–Sternberg cells with the CD30+/CD15+ phenotype, which are not observed in LyP [100]. A table of possible differential diagnoses of LyP is presented below, taking into account their similarities and differences for each of the subtypes of this disease (Table 2).

Differential diagnosis of LyP should also include inflammatory and infectious conditions that contain a significant number of CD30+ cells and mimic the pathological and clinical disease in question. Such conditions include pityriasis lichenoides et varioliformis acuta (PLEVA), reactions to arthropod bites or nodular scabies, drug-induced lymphatic eruptions or viral infection (Figure 3). The Kempf team’s study on the differentiation between PLEVA and LyP showed that the two disease subunits may share not only histopathological or clinical but also molecular characteristics. The exception is the lymphocytic infiltration described by the researchers, which is characterized in PLEVA by the CD8+/CD30+ phenotype [101]. 

An interview for chronic reactions to insect bites or the occurrence of nodular scabies, which are clinically similar to LyP, seems to be important in the diagnosis. A history of exposure and the presence of symptoms (e.g., intense itching) can help distinguish arthropod bites from LyP. Additionally, a detailed analysis of skin lesions should reveal the presence of a dense inflammatory infiltrate composed of lymphoid cells and histiocytes with an admixture of eosinophils and plasma cells and atypical mononuclear cells with hyperchromatic nuclei [102]. Skin lesions in the form of lymphatic eruptions may resemble LyP, both clinically and histopathologically. Hardened lumps or plaques resulting from the use of antibiotics, antiepileptic drugs, or biologics may be characterized by large, atypical CD30+ T cells [103,104,105]. A study by the team of Leinweber et al. in 2006 revealed that clinical and histopathological symptoms similar to LyP are also observed in the course of some viral infections. This mainly concerns herpes simplex, varicella zoster, and molluscum contagiosum infections. The course of the infection shows large atypical CD30+ cells in histological examination, but the accompanying skin lesions do not disappear, and there is no rearrangement of the T cell receptor gene [106].

## 7. Possible Treatment Strategies

Due to the recurrent and chronic nature of LyP, the treatment used is symptomatic and is aimed at accelerating the healing of lesions or reducing their severity. Therefore, the first strategy for treating LyP is to observe the patient’s condition and changes. This applies to both patients with limited or few asymptomatic lesions (without scarring), as well as patients with extensive and symptomatic disease. It depends largely on the preferences of the patient and the degree of their coping with the disease [107]. There are suggestions in the literature that therapeutic strategies selected by doctors differ depending on the location, severity, and extent of the lesions, but not on the type of LyP (Figure 4A,B) [62].

To date, there are four main and widely practiced therapeutic approaches: phototherapy, topical steroid use, low dose methotrexate, and an alternative therapeutic route including antibiotics, antihistamines, or oral steroids [62,108]. All these therapeutic approaches can be classified as first-line treatment after standard patient observation. In the literature, we can find various studies on the effectiveness of the above-mentioned therapies. However, the work of Fernández-de-Misa et al. in 2017 deserves special attention. This team conducted a retrospective analysis of the efficacy of these therapies as first-line treatment [108]. The analysis used 252 patients, 193 of whom were treated with standard therapeutic approaches; 34.52% of patients were treated with topical steroid drugs, 20.24% of patients were treated with low dose methotrexate, 14.29% of patients were treated with phototherapy, and 7.54% received an alternative treatment route. The remaining 23.41% of patients were in the control group and did not receive any of the above-mentioned therapies. The investigators measured the response of treated patients to the treatment given by the development or disappearance of skin lesions. Complete response to treatment was defined as resolution of all active lesions without developing new lesions (CRs), a partial response was defined as regression of at least 50% of active lesions and development of fewer new lesions (PR), and others were defined as no response (NR). The overall response to treatment among all analyzed patients was only 48% (44% of patients using topical steroids; 52% of patients using methotrexate; 61% of patients receiving phototherapy; 37% of patients receiving alternative therapy). The researchers noted that there was no statistically significant difference in the choice of treatment therapy with regard to the patient’s gender or the extent of the skin disease. Additionally, their analysis established that the estimated median time to CR was about 10 months and that there were no significant occurrences between the analyzed treatments. Moreover, the researchers showed that of 86 patients with CR, 78% of them had a cutaneous recurrence; this proportion was similar for all analyzed treatments [108].

LyP requires constant monitoring of the patient’s lesions and usually ends with the administration of topical steroids as initial treatment. Topical steroids can be used alone (Figure 5) or in combination with other drugs; however, none of the methods mentioned prevent the development of new lesions and relieve symptoms relatively quickly. Research indicates that oral corticosteroids are ineffective [109].

Another part of the management strategy involves the use of phototherapy (most often PUVA). Data from the literature indicate that PUVA should be administered twice a week for six to eight weeks or until symptoms resolve [107]. There are several studies showing the effectiveness of treating LyP patients with this type of therapy. It has been shown that its use allowed for significantly higher disease-free survival rates [108,110,111]. In a review by Kempf et al. in 2011, it was shown that CR after phototherapy in LyP patients was observed only in 26.32% of patients, and PR in 68.42% of patients [62]. Although this method has been reported to significantly reduce the number of skin lesions, its long-term use may induce skin carcinogenesis (especially melanoma) [112,113].

The most widely used chemotherapeutic agent in the treatment of LyP is methotrexate, which has been reported to be very effective in controlling the disease, but due to frequent relapses, its use should be extended, which is also associated with an increased risk of side effects [114,115,116]. According to the recommendations of the Dutch Cutaneous Lymphoma Group (2015), the initial dose of methotrexate should be 7.5–10 mg once a week in combination with folic acid supplementation (5 mg) [117]. In the literature, we found several studies on its effectiveness, which are presented in Table 3.

When treating patients with methotrexate, particular attention should be paid to the complications and side effects that occur in the case of long-term use of the drug. According to the literature, the majority of patients treated with low doses of methotrexate experience side effects such as nausea, stomach upset, headache, and fatigue [120]. Doctors recommend that patients be screened for hepatitis B and C infection before starting long-term treatment with methotrexate due to the drug’s hepatotoxicity. Monitoring of serum transaminases and peripheral blood counts is also recommended during treatment, which should be performed twice a month for the first month and then every 4–12 weeks for liver damage [120,121].

Apart from the methods of treatment of LyP described above, there are examples of other therapies in the literature which, due to a small trial, require further research on their effectiveness (Figure 6) [40,109,116,122,123,124,125,126,127,128,129].

The main clinical challenge remains the frequent relapses that persist even after successful first stages of treatment. Clinicians should also carefully consider their selected therapies because, for example, multicomponent chemotherapy can cause serious complications in both the short-term and long-term.

## 8. Materials and Methods

### Search Strategy, Study Selection, and Data Extraction

The literature search was carried out on the PubMed and Web of Science databases, where the search for available articles was performed based on the following keywords: “lymphoproliferative disorder”, “LyP”, and “immune system”. No specific timeframe, geographical scope, or language restrictions were applied. Duplicates were rejected from the list of identified articles. The suitability for the inclusion of each work into the analysis was thoroughly assessed. Outside of the established inclusion criteria for this review, in the course of evaluation, 11 additional publications deviating from the initial timeframe were also included. Eventually, 141 articles were included in the review.

## 9. Conclusions

LyP remains a mysterious and poorly understood condition. It is clinically important to recognize that LyP is capable of mimicking several other diseases and therefore requires histopathological evaluation with immunophenotypic and genetic examination to rule out other malignancies, avoid possible misdiagnosis, and accelerate treatment progress. Although LyP mainly affects adults in the fourth and fifth decade of life, it has also been reported to develop in children as young as 1 year old.

Due to the increased risk of lymphoma development, patients diagnosed with LyP require lifelong follow-up, and the vast majority will develop malignant neoplasms within 20–30 years. Current treatment options allow effective disease control, but careful observation while refraining from actively treating the disease is a viable option for localized or mild cases.

## Figures and Tables

**Figure 1 cells-11-03697-f001:**
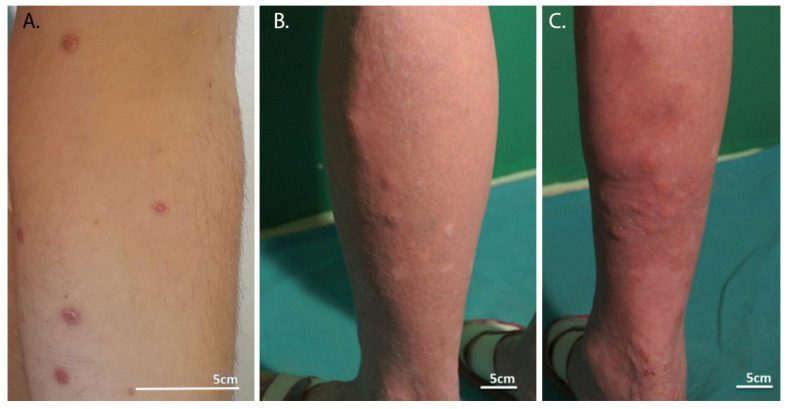
Skin lesions in lymphomatoid papulosis. (**A**) Widespread erythematous papulonodular eruptions with scaling of the lesions (reproduced from Kavvalou et al. [45]). (**B**) Typical manifestation of lymphoid clumps (authors’ archive). (**C**) Typical manifestation with clustered nodules on the patient’s skin (authors’ archive).

**Figure 2 cells-11-03697-f002:**
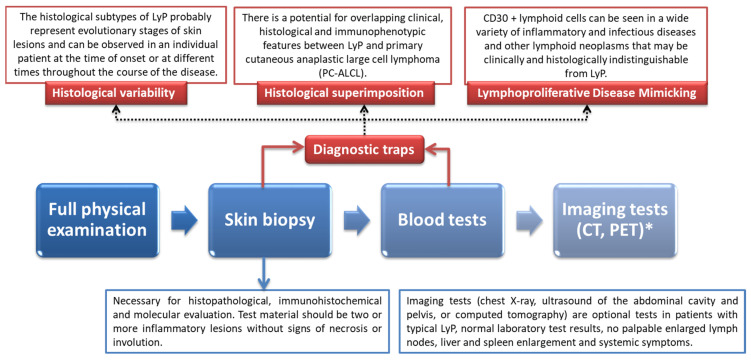
Proper diagnostic procedure in case of suspected lymphomatoid papulosis.

**Figure 3 cells-11-03697-f003:**
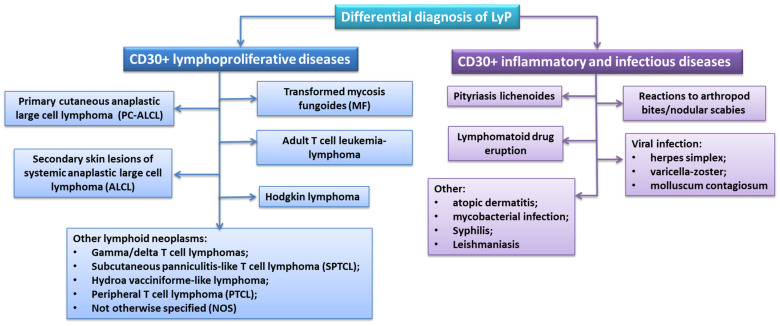
Differential diagnosis of LyP.

**Figure 4 cells-11-03697-f004:**
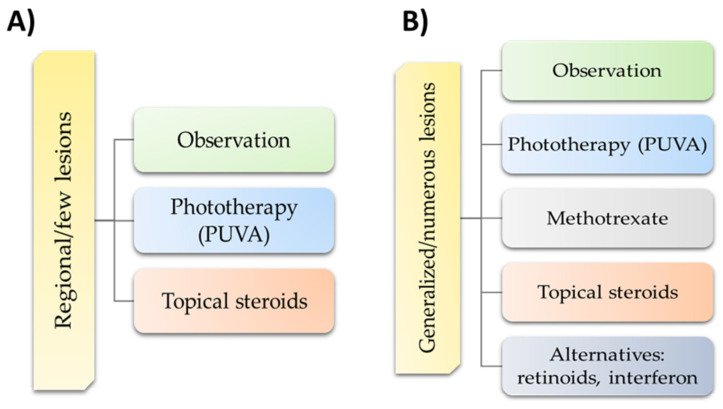
Management of regional lymphomatoid papulosis according to EORTC, ISCL, and USCLC consensus recommendations. (**A**) Management of regional lymphomatoid papulosis with few lesions according to EORTC, ISCL, and USCLC consensus recommendations; (**B**) management of generalized lymphomatoid papulosis with numerous lesions according to EORTC, ISCL, and USCLC consensus recommendations. Abbreviations: EORTC—European Organisation for Research and Treatment of Cancer; ISCL—International Society for Cutaneous Lymphomas; USCLC—United States Cutaneous Lymphoma Consortium; PUVA—Psoralen Ultra-Violet A.

**Figure 5 cells-11-03697-f005:**
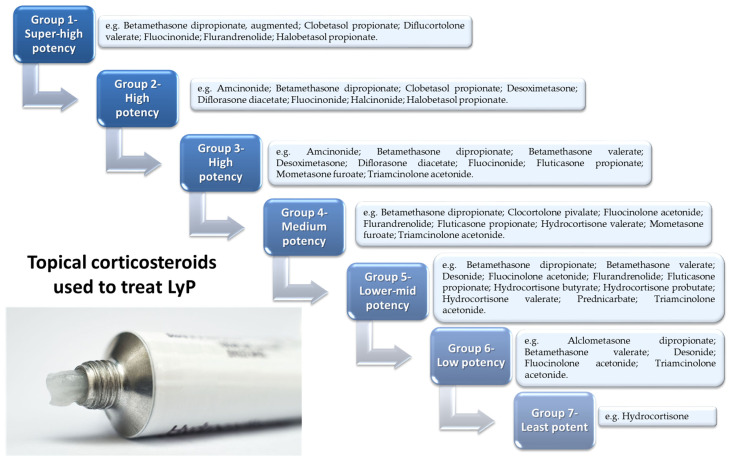
The most common topical corticosteroids for treatment of LyP, listed by potency according to the United States classification system: group 1 is the most potent, group 7 is the least potent. The degree depends on the concentration of the corticosteroid used, which decreases as the group number increases.

**Figure 6 cells-11-03697-f006:**
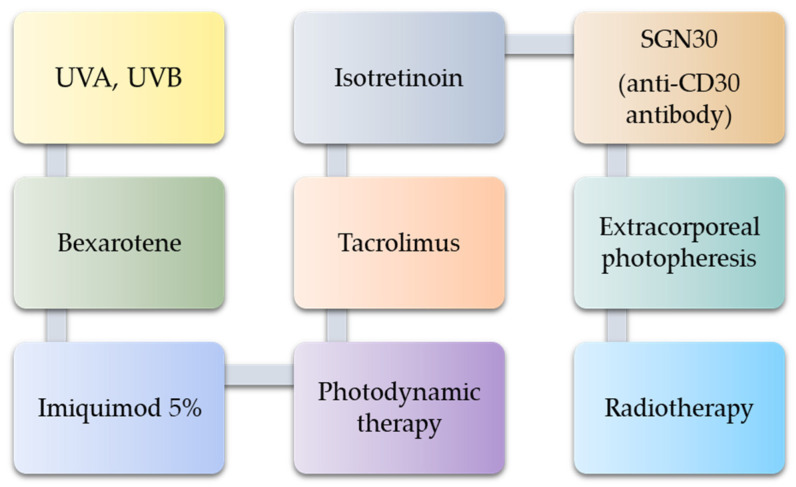
Less-frequent therapeutic strategies for lymphomatoid papulosis treatment.

**Table 1 cells-11-03697-t001:** Participation of individual CD markers in immunohistochemical analyses of lymphomatoid papulosis types.

Type	Cluster of Differentiation								
	CD30	CD2	CD3	CD4	CD5	CD7	CD8	TIA1	References
Type A LyP	+	+/−	+	+	+/−	+/−	−	+	[11]
Type B LyP	+/−	+/−	+	+	+/−	−	−	+	[11]
Type C LyP	+	+/−	+	+	+/−	+/−	−	+	[11]
Type D LyP	+	+	+	−	−	−	+	+	[63]
Type E LyP	+	+	+	+/−	+	+/−	+	+	[63]
Type with rearrangement 6p25.3	+	+/−	+	+	+/−	+/−	+/−	−	[63]

Abbreviations: (+) indicates expression of the given CD marker; (−) indicates no expression of a given CD marker; (+/−) indicates the variable expression of the given CD marker. LyP: lymphomatoid papulosis; TIA1: T cell intracellular antigen 1.

**Table 2 cells-11-03697-t002:** Possible differential diagnoses of lymphomatoid papulosis.

Differential Diagnosis				
Lymphomatoid Papulosis	Main Differential Diagnosis	Similarities	Differences	Reference
Each subtype	Primary cutaneous ALCL	CD30+ expression Good prognosis Primary lymphoproliferation	Clinical outcome: LyP typically represents spontaneously regressing lesions, primary cutaneous ALCL represents solitary, solid, ulcerated nodules	[75,90]
Type A	Classic Hodgkin’s lymphoma, Primary cutaneous ALCL, tumor stage MF	Same immunophenotype of large cells and primary cutaneous ALCL malignant cells MF cells also may express CD30+	Lack of T cell antigen expression by LyP Reed–Stenberg-like cells	[75,90,91]
Type B	Plaque stage MF	MF cells also can express CD30+	-	[75]
Type C	Primary cutaneous ALCL, transformed MF	Same immunophenotype of large cells and ALCL malignant cells MF cells also can al express CD30+	Transformed MF has an aggressive clinical course	[65,75]
Type D	PCAETL resembling pagetoid reticulosis	Cytotoxic phenotype of cells (CD8 and granzyme co-expression)	Aggressive epidermotropic cytotoxic T-cell lymphoma cells rarely express CD30+ Lymphoma has a more aggressive clinical outcome	[61,76]
Type E	NK/T-cell lymphoma nasal type, extranodal CGD-TCL ALCL/borderline CD30+ cutaneous LPD	Clinically ulcerative lesions leaving scars NK/T-cell lymphoma histologically present angioinvasion, angiocentricity, necrosis In CGD-TCL, more than one histological subtype presented at one time In CGD-TCL, lymphoma angioinvasion Cytotoxic phenotype CD30+ cells in LyP type E, ALCL, borderline LPD	Typically lack CD30+ expression in NK/T cell lymphoma NK/T cell lymphoma is EBV-associated In CGD-TCL, lack of CD4 expression and variable CD8 expression TCR expression in gamma/delta T-cell lymphomas Aggressive clinical outcome in extranodal lymphomas Poor prognosis in extranodal lymphomas Rarely described angiodestructive potential for infiltrating cells in borderline LPD and ALCL	[70,92,93,94,95,96,97]
Type with 6p25.3 rearrangement	ALCL	Up to 50% of primary cutaneous ALCL present the same rearrangement	-	[61,77]

Abbreviations: ALCL—anaplastic large cell lymphoma; MF—transformed mycosis fungoides; PCAETL—primary cutaneous aggressive epidermotropic cytotoxic T-cell lymphoma; CGD-TCL—gamma/delta T-cell lymphoma-extranodal; LPD—lymphoproliferative disorders.

**Table 3 cells-11-03697-t003:** Treatment efficacy studies with methotrexate in patients with LyP.

The Authors of the Study	Description of the Research and Its Results	Reference
Vonderheid et al.	45 LyP patients treated with methotrexate 15–25 mg per week 20 patients (44 percent) did not develop new lesions, and 19 patients (42 percent) developed only a few lesions during treatment After treatment discontinuation, 10 out of 40 patients had no relapse after a follow-up of 24 to 227 months Side effects were reported in 77 percent of patients, including liver fibrosis in 5 out of 10 patients treated with methotrexate for more than three years.	[118]
Bruijn et al.	28 adult LyP patients were treated with oral methotrexate 5 to 25 mg per week for 1 to 216 months (median 37 months) Transient increases in liver enzymes occurred in 10 patients during the first month of treatment 2 patients discontinued treatment due to persistently elevated levels of liver enzymes	[117]
Newland et al.	25 patients with LyP were treated with oral methotrexate 20–30 mg per week for at least six months and then for two to six months during the withdrawal period 22 patients had a partial or complete response Only 6 successfully stopped the drug and maintained a response for six months 16 patients remained addicted to methotrexate	[119]

## Data Availability

Not applicable.

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
