# Peer review of "Etiopathogenesis, Diagnosis, and Treatment Strategies for Lymphomatoid Papulosis with Particular Emphasis on the Role of the Immune System"

_cells, 2022, doi:10.3390/cells11223697_

Round 1
Reviewer 1 Report
This review by Nowicka et al. is well worth publishing, as there have not been many review articles concerning lymphomatoid papulosis.
However, I have some concerns before accepting this manuscript.
Section 4 (Etiopathogenesis and role of the immune system in the development of LyP) is too redundant.
Certainly, the etiology of LyP is largely unknown and little can be explained in this paragraph. However, the author has written much about CTCL in general in this paragraph. This paragraph should be much shorter and restricted only to what is involved in the etiology of LyP.
Figure 1
The clinical manifestation of LyP is characterized by small papules, as described by the authors.
At least Figures 1B and 1C are not the typical clinical picture of LyP.
I do not consider them (especially figure 1C) appropriate to be included as clinical images of LyP in a review article.
Figure 1A is also too highly magnified to be appropriate for presentation in a review article. Figures 1(b) and 4(b) of Ref. 43 are more appropriate if the figures are taken from Ref. 43.
The authors mentioned that the time range of the searched articles was established for the years 2000 to 2022 in section 8.1 (page 11, lenes 437 – 438), while they also cited several papers published prior to 2000 in this paper.
The address on the website in reference 18 is incorrect.
The correct address is https://www.ncbi.nlm.nih.gov/books/NBK532295/
Author Response
Dear Reviewer,
Thank you for commenting on our paper, please find below all the answers for your concerns.
This review by Nowicka et al. is well worth publishing, as there have not been many review articles concerning lymphomatoid papulosis.
Thank you for taking the time to review the manuscript and for providing insightful comments that helped us to improve the manuscript.
However, I have some concerns before accepting this manuscript.
Section 4 (Etiopathogenesis and role of the immune system in the development of LyP) is too redundant.
Certainly, the etiology of LyP is largely unknown and little can be explained in this paragraph. However, the author has written much about CTCL in general in this paragraph. This paragraph should be much shorter and restricted only to what is involved in the etiology of LyP.
Thank you for this comment. We have added an explanation why we focus on etiology of CTCL and shortened this chapter.
Figure 1. The clinical manifestation of LyP is characterized by small papules, as described by the authors. At least Figures 1B and 1C are not the typical clinical picture of LyP. I do not consider them (especially figure 1C) appropriate to be included as clinical images of LyP in a review article. Figure 1A is also too highly magnified to be appropriate for presentation in a review article. Figures 1(b) and 4(b) of Ref. 43 are more appropriate if the figures are taken from Ref. 43.
We agree with the reviewer. We have replaced the figures in question with others from our own archives.
The authors mentioned that the time range of the searched articles was established for the years 2000 to 2022 in section 8.1 (page 11, lenes 437 – 438), while they also cited several papers published prior to 2000 in this paper.
Thank you for spotting this mistake. We have corrected the timeframe of our search.
The address on the website in reference 18 is incorrect. The correct address is https://www.ncbi.nlm.nih.gov/books/NBK532295/
Thank you for your scrutiny. This has been corrected.
Again, thank you for reading our paper and making suggestions, that impacted (as we hope) its improvement.
We do hope that in the current revised form the paper will match the requirements.
Your sincerely,
Paulina Niedźwiedzka-Rystwej
Reviewer 2 Report
Nowicka et al embarked in a review on lymphomatoid papulosis bringing an extensive amount of informations which are of limited help to the clinician addressing a patient with the disease. The review should be entirely rewritten and particularly the following suggestions should be taken into consideration:
1. Introduction should be shortened and more concise on the topic (why mentioning LPD, of course all know that we are in the field of LPD. I'm definitely up to see a more straight introduction
2. Table 2 is confusing. I would try to design it by placing three row (clinical-histopatology-genetics) and all columns with each differential diagnosis (Lyp, ALCL, MF and other benign conditions CD30+)
3. I would definitely add a comprehensive chapter discussing differential diagnosis, which is rather wide.
4. Treatment stategies chapter should be entirely rewritten considering the therapeutic approach based on extension of the disease, first line, second line, evidence from the literature. I suggest to consider a chapter helpful to the clinician who is in the process of choosing the more appropriate treatment
5.
| Clinical | Self-healing papulonodular skin lesions | Frequent secondary skin lesions, lymphadenopathy B symptoms | Scaling erythematous patches, plaques, tumors, +/– lymphadenopathy | Scaling erythematous hemorrhagic lesions | Exposure |
Itchy lesions Responds well to scabies treatment |
| Histopathology/ immunophenotype | Atypical CD30, CD4+ cells surrounded by inflammatory cells |
Lack of epidermotropic cerebriform cells ALK+, EMA+; ALK– in patients over 30 |
Epidermotropism of cerebriform cells CD30+ with large cell transformation |
Interface dermatitis, necrotic keratinocytes, extravasated erythrocytes CD8>CD4, few CD30+ cells |
Punctum, insect parts Polymorphous Inflammation CD30+ cells may be present |
Presence of mite CD30+ cells and B cells present |
| Genetics |
Clonal in 60% Aneuploidy of type A Lack of t(2;5) Recurrent rearrangement of 6p25.3
|
t(2;5) often present No IRF4 translocation Clonal TCRrearrangement |
Lack of t(2;5) Complex karyotype, clonal or oligoclonal TCRrearrangement |
Clonal TCRrearrangement in 50 percent | No abnormalities | No abnormalities |
Author Response
Dear Reviewer,
Thank you for commenting on our paper, please find below all the answers for your concerns.
Nowicka et al embarked in a review on lymphomatoid papulosis bringing an extensive amount of informations which are of limited help to the clinician addressing a patient with the disease. The review should be entirely rewritten and particularly the following suggestions should be taken into consideration:
Thank you for allowing our paper to be reviewed and for the consequent opportunity to further improvement. We are very grateful for the time you spent on reading and commenting our manuscript. We hope that the approach we have taken improved its value.
- Introduction should be shortened and more concise on the topic (why mentioning LPD, of course all know that we are in the field of LPD. I'm definitely up to see a more straight introduction
The introduction has been redrafted and shortened as suggested
- Table 2 is confusing. I would try to design it by placing three row (clinical-histopatology-genetics) and all columns with each differential diagnosis (Lyp, ALCL, MF and other benign conditions CD30+)
Than you for this comment. The rationale behind the structure of this table is show differences and similarities between types of lymphomatoid papulosis and other pathologies based on data from other publications. We are afraid that when we change the structure of this table, such clear differentiation.
- I would definitely add a comprehensive chapter discussing differential diagnosis, which is rather wide.
Thank you for your suggestion, as recommended, we have added subsection 6.1 for differential diagnosis.
- Treatment stategies chapter should be entirely rewritten considering the therapeutic approach based on extension of the disease, first line, second line, evidence from the literature. I suggest to consider a chapter helpful to the clinician who is in the process of choosing the more appropriate treatment
This section has been completely rewritten, we have discussed each of the commonly used treatment strategies, we hope that the prepared description will allow doctors to choose the appropriate therapeutic strategy.
|
Clinical |
Self-healing papulonodular skin lesions |
Frequent secondary skin lesions, lymphadenopathy B symptoms |
Scaling erythematous patches, plaques, tumors, +/– lymphadenopathy |
Scaling erythematous hemorrhagic lesions |
Exposure |
Itchy lesions Responds well to scabies treatment |
|
Histopathology/ immunophenotype |
Atypical CD30, CD4+ cells surrounded by inflammatory cells |
Lack of epidermotropic cerebriform cells ALK+, EMA+; ALK– in patients over 30 |
Epidermotropism of cerebriform cells CD30+ with large cell transformation |
Interface dermatitis, necrotic keratinocytes, extravasated erythrocytes CD8>CD4, few CD30+ cells |
Punctum, insect parts Polymorphous Inflammation CD30+ cells may be present |
Presence of mite CD30+ cells and B cells present |
|
Genetics |
Clonal in 60% Aneuploidy of type A Lack of t(2;5) Recurrent rearrangement of 6p25.3
|
t(2;5) often present No IRF4 translocation Clonal TCRrearrangement |
Lack of t(2;5) Complex karyotype, clonal or oligoclonal TCRrearrangement |
Clonal TCRrearrangement in 50 percent |
No abnormalities |
No abnormalities |
Again, thank you for reading our paper and making suggestions, that impacted (as we hope) its improvement.
We do hope that in the current revised form the paper will match the requirements.
Your sincerely,
Paulina Niedźwiedzka-Rystwej
Reviewer 3 Report
In general, the manuscript is well written. However, this manuscript is not good for scope of cells. The mechanism should be addressed more.
Author Response
Dear Reviewer,
Thank you for commenting on our paper, please find below all the answers for your concerns.
In general, the manuscript is well written. However, this manuscript is not good for scope of cells. The mechanism should be addressed more.
Thank you for taking the time to read and evaluate the manuscript. We are glad that you find the paper interesting and well-written. This review presents a comprehensive approach to summarising the available evidence on lymphomatoid papulosis. There are not many review articles on this pathology, so we think this review fill the important information gap. However, we agree that mechanisms underlying lymphomatoid papulosis and its etiopathogenesis is not well understood. Such reviews may encourage further research in this area, and for this reason they may be of value and worth publishing.
Again, thank you for reading our paper and making suggestions, that impacted (as we hope) its improvement.
We do hope that in the current revised form the paper will match the requirements.
Your sincerely,
Paulina Niedźwiedzka-Rystwej
Round 2
Reviewer 2 Report
The revised version has been substantially improved and I believe suitable for publication
Reviewer 3 Report
This manuscript has been sufficiently revised according to the opinions of reviewers.
Finally, the revised paper is ready for publication in Cells.